# Direct Interaction Between CD34^+^ Hematopoietic Stem Cells and Mesenchymal Stem Cells Reciprocally Preserves Stemness

**DOI:** 10.3390/cancers16233972

**Published:** 2024-11-27

**Authors:** Rémi Safi, Tala Mohsen-Kanson, Farah Kouzi, Jamal El-Saghir, Vera Dermesrobian, Inés Zugasti, Kazem Zibara, Pablo Menéndez, Hiba El Hajj, Marwan El-Sabban

**Affiliations:** 1Department of Anatomy, Cell Biology and Physiological Sciences, American University of Beirut, Beirut 1107, Lebanon; rsafi@carrerasresearch.org (R.S.); ejamal@med.umich.edu (J.E.-S.); vera.dermesrobian@kuleuven.be (V.D.); 2Josep Carreras Leukemia Research Institute, 08916 Barcelona, Spain; pmenendez@carrerasresearch.org; 3Faculty of Science, Lebanese University, Zahle 1801, Lebanon; tala.kanson@ul.edu.lb (T.M.-K.); farah.kouzi@lih.lu (F.K.); kzibara@ul.edu.lb (K.Z.); 4Faculty of Science, Lebanese University, Hadath 40016, Lebanon; 5Division of Nephrology, Department of Internal Medicine, University of Michigan, Ann Arbor, MI 48109, USA; 6Laboratory of Adaptive Immunity, Department of Microbiology, Immunology and Transplantation, KU Leuven, 3000 Leuven, Belgium; 7Department of Hematology, Hospital Clínic Barcelona, 08036 Barcelona, Spain; izugasti@clinic.cat; 8Institució Catalana de Recerca i Estudis Avançats (ICREA), 08010 Barcelona, Spain; 9Consorcio Investigación Biomédica en Red de Cancer, CIBER-ONC, ISCIII, 28029 Barcelona, Spain; 10Spanish Network for Advanced Cell Therapies (TERAV), 08028 Barcelona, Spain; 11Department of Experimental Pathology, Immunology and Microbiology, Faculty of Medicine, American University of Beirut, Beirut 1107, Lebanon; he21@aub.edu.lb

**Keywords:** bone marrow, microenvironment, CD34^+^ hematopoietic stem cells, mesenchymal stem cells, connexin 43, N-Cadherin, gap junction, acute myeloid leukemia, heterocellular interaction

## Abstract

The bone marrow microenvironment regulates the fate of hematopoietic cells through direct or indirect communication. Although the paracrine interaction of the cellular components of the niche has been extensively investigated, little is known about the direct communication between mesenchymal stem cells (MSCs) and hematopoietic stem cells (HSCs). In this study, we investigated the reciprocal direct interaction between MSCs and CD34^+^ HSCs under physiological conditions and hematological malignancies such as acute myeloid leukemia (AML). Our results demonstrated that cell–cell communication between MSCs and CD34^+^ HSCs is mediated through junctional complexes (connexin 43 and N-Cadherin interaction), preserving the stemness of MSCs. Interestingly, communication is reduced in AML, presumably due to a pathological defect in MSCs, exacerbating leukemia progression.

## 1. Introduction

Hematopoietic stem cells (HSCs) and their progenies, which give rise to all blood cells, reside in a special microenvironment in the bone marrow (BM) [1]. Under physiological conditions, most HSCs are quiescent, and only a small fraction of HSCs enters the cell cycle and differentiates into multipotent progenitors [2,3,4,5,6]. Intrinsic or extrinsic signals induce the differentiation of these progenitors to common lymphoid and myeloid progenitors under the so-called process of hematopoiesis [7]. In hematological malignancies, aberrations in the bone marrow microenvironment limit the regeneration and differentiation potential of HSCs by reducing their numbers and/or function [8]. Therefore, the hematopoietic system is well-organized in a hierarchical differentiation cascade, where each phase in the differentiation process requires a different combination of cytokines, chemokines, extracellular matrix (ECM) interactions, and cell–cell interactions, which are offered by different stromal cell types [9]. Mesenchymal stem cells (MSCs) are the pivotal stromal cellular players of the endosteal niche. They are in intimate interaction with HSCs, maintaining their stem cell quiescence, self-renewal, apoptosis, and differentiation properties through the production of cytokines, chemokines, and ECM [3]. Several studies have shown that the indirect interaction between MSCs and HSCs is maintained by the secretion of thrombopoietin (TPO), stem cell factor (SCF), and fms-like tyrosine kinase 3/fetal liver kinase-2 (FLT3/Flk2) cytokines that regulate the function of HSCs [10,11]. Moreover, MSCs possess C-X-C motif chemokine 12/stromal-derived factor 1 (CXCL12/SDF1), SCF/c-Kit and Slit-2/Robo-4 chemokines, which play a role in the trafficking and homing of HSCs [2,4,5,12,13]. In addition to the production of a large array of regulatory molecules, recent studies document the role of MSCs in affecting the function of HSCs through direct contact [7,14]. Communication of cells by cellular contacts could be mediated through membrane proteins, forming gap junctions, that facilitate exchanges of small molecules (<2 KDa), secondary messengers, and ions between various cells, for example, Ca^2+^, cyclic adenosine monophosphate (cAMP), and inositol triphosphate (IP3) [15]. Cell–cell communication through gap junctions between HSCs and MSCs or stromal cells in the hematopoietic microenvironment of the human system is sparse. Yet, functional connexin (Cx)-43 gap junctions exist between stromal cells and immature hematopoietic progenitor cells in humans [16], and Cx-43 and Cx-37 mRNA are expressed at low levels in human BM and cord blood hematopoietic progenitor CD34^+^ cells [15,17]. This provides evidence of an important regulatory pathway of hematopoiesis through gap junctions in humans. Studies have shown that cadherins are major cell adhesion molecules responsible for Ca^2+^-dependent cell–cell interaction [18,19]. The association between gap junctions and adherens junctions, particularly between Cx-43 and N-Cadherin (N-Cad), is required for gap junction assembly and function [20]. The role of N-Cad in HSCs-BM niche interactions is debatable. While some studies showed that N-Cad conditional knockout mice had no observable phenotype in HSCs maintenance and hematopoiesis [21], others demonstrated that the inhibition of N-Cad expression reduced the anchoring of HSCs to the endosteal surface and inhibited their long-term engraftment [22].

Acute myeloid leukemia (AML) is a heterogeneous clonal disorder characterized by an increase in the number of leucoblast cells featuring cytogenetic aberrations and recurrent somatic mutations [23,24]. Intercellular communication with stromal cells affects AML cell proliferation, and apoptosis [25], and increases their drug resistance [26], thus promoting AML relapse [27]. Weber et al. demonstrated that the direct interaction of human myeloid leukemia cell lines with BM stromal cells decreased leukemic cell differentiation [28]. Recently, a study showed that gap junction inhibition reduced the chemoresistance to cytarabine, used as a standard-of-care chemotherapy in AML [17]. Moreover, inhibiting gap junction formation between AML cells and stromal cells using carbenoxolone diminished AML chemoresistance, which is triggered by BM-MSCs [28].

Although the role of soluble mediators was extensively studied in the differentiation process of HSCs and in the promotion of leukemic cells [29,30], very little is known about the role of direct communication via gap junctions in modulating these two processes. In this study, we investigated the role of MSCs in modulating HSCs differentiation, via direct interactions through N-Cad and Cx-43 under physiological conditions. We also examined whether the cell–cell interaction between MSCs and AML cells is differentially regulated under pathological conditions such as AML.

## 2. Materials and Methods

### 2.1. Isolation and Phenotypic Identification of CD34^+^ Cells

Since we used leftover bone marrow aspirates with no identifiers and mobilized peripheral cells from deceased patients, the study protocol is deemed non-human research and hence exempt from approvals by the Institutional Review Board of the American University of Beirut. Frozen mobilized peripheral blood. Blood was layered on Ficoll-Paque (Sigma-Aldrich, Burlington, MA, USA) density gradient and centrifuged at 400× *g* for 30 min. Mononuclear cells were recuperated, washed, and stained with different antibodies.

FITC mouse anti-CD38, APC mouse anti-CD45, and PE anti-CD34 (BD, Franklin Lakes, NJ, USA) were used for separate immunophenotyping of cells. The isotype control antibodies were either FITC mouse IgG2a or APC mouse IgG1 (BD, Franklin Lakes, NJ, USA). For each staining, 10,000 events were acquired by flow cytometry.

For isolation of CD34^+^ cells, cells with small size and granularity were selected (according to HSC size less than 10 µm [31] and further analyzed for the expression of the CD34 marker. CD34^+^ cells were sorted using FACS Aria SORP cell sorter (BD, Franklin Lakes, NJ, USA). The purity of isolated CD34^+^ cells typically ranged between 90 and 99%. The percentage of CD34^+^ cells varied between 0.5 and 4% among patients diagnosed with T-cell Lymphoma, Hodgkin Lymphoma, Non-Hodgkin Lymphoma, Lymphoma, Multiple Myeloma, Ewing Sarcoma, and Neuroblastoma. Moreover, immunophenotypic characterization of differentiation markers indicated that these cells were an enriched population of HSCs with high self-renewing capacity since at least 50% of CD34^+^ cells lack the expression of CD45 and CD38 markers (Appendix A). In this study, we focused on CD34^+^ cells isolated from Multiple Myeloma and Hodgkin Lymphoma patients, since the mutations causing the disease do not affect the CD34 progenitors [32].

CD34^+^ HSCs were sorted from the whole MNC population, and an enriched population of 93% of CD34^+^ cells was obtained (Appendix A). Colony-forming assay (CFU) was performed on CD34^+^ cells to assess their clonogenic potential. Results showed that these cells were able to form all types of colonies including granulocyte (CFU-G), macrophage (CFU-M), granulocyte–macrophage (CFU-GM), erythroid (CFU-E), granulocyte–erythrocyte–macrophage–megakaryocyte (CFU-GEMM) in human methylcellulose complete medium (R&D systems, Minneapolis, MN, USA) (Appendix A).

### 2.2. Co-Culture System of MSCs with HSCs or with AML Cells

Normal healthy MSCs were purchased from Lonza (Basel, Switzerland) or isolated from leftover aspirates from donors and AML patients. Two AML cell lines, THP-1 and Molm-13, were grown routinely in RPMI medium supplemented with 10% fetal bovine serum and 1% penicillin–streptomycin. Three primary cells isolated from AML patients, having different genetic and cytogenetic backgrounds (Appendix A) were used in this study after approval from the Institutional Review Board of Clinic Hospital in Barcelona. AML primary cells were grown in Stem Spam medium (STEMCELL Technologies, Saint Égrève, France) supplemented FLT3 (90 ng/mL), IL-3 (10 ng/mL), and SCF (90 ng/mL). MSCs were seeded (7000/cm^2^) in 6-well plates in DMEM low-glucose medium (Sigma-Aldrich, Burlington, MA, USA). After reaching 80% confluency, cells were switched to IMDM medium 24 h prior to co-culture. Freshly isolated CD34^+^ cells or AML cell lines were directly co-cultured with MSCs (ratio 1:1). After 24 h, two CD34^+^ cell populations and AML cell lines were identified: adherent and suspension fractions. CD34^+^ HSCs or AML cell lines in suspension were collected, and then the adherent fraction was separated from MSCs by gentle PBS washes. The purity of this separation between MSCs and HSCs was verified by FACS analysis according to the size and expression of a specific mesenchymal stem cell marker CD73 [14], and by showing that the HSCs adherent fraction is smaller than MSCs and more than 95% of these small cells do not express CD73 mesenchymal marker (Appendix A).

### 2.3. Colony Forming Cell Assay

CD34^+^ cells (1 × 10^3^) were cultured for 15 days in human methylcellulose complete medium, which includes the cytokines SCF (50 ng/mL), GM-CSF (10 ng/mL), IL-3 (10 ng/mL), and EPO (3 IU/mL). Colonies of erythroid progenitors (CFU-E and BFU-E), granulocyte–macrophage progenitors (CFU-GM, CFU-G, and CFU-M), and multi-potential granulocyte–erythroid–macrophage–megakaryocyte progenitors (CFU-GEMM) were morphologically characterized, counted, and quantified by light microscopy (Zeiss, Oberkochen, Germany).

### 2.4. Transcriptional Expression Analysis

Total RNA was extracted from cells using RNeasy Mini Kit (Qiagen, Germantown, MD, USA) as per manufacturer’s instructions. Briefly, 1 μg of total RNA was reverse transcribed to cDNA using RevertAid 1st Strand cDNA Synthesis Kit (Thermo Scientific, Lenexa, KS, USA). Quantitative PCR (qPCR) was performed using the iQ SYBR GreenSupermix in a CFX96 system (Bio-Rad Laboratories, Hercules, CA, USA). Products were amplified using primers that recognize Cx-43, N-Cad, MMP2, SDF-1, VEGF, Snail, Twist, Nanog, Oct-4, CXCR4, ALP, and GAPDH (Table 1). PCR parameters consist of a pre-cycle of 95 °C for 3 min followed by 40 cycles consisting of 95 °C for 10 s, 52–62 °C for 30 s, and 72 °C for 30 s with a final extension at 72 °C for 5 min. The fluorescence quantitative cycle value (Cq) was obtained for each gene and normalized to their corresponding GAPDH in the same sample. All experiments were carried out in duplicates and independently performed three times.

### 2.5. Protein Expression Analysis

Western blot analysis was performed to examine the levels of Cx-43 and N-Cad expression in MSCs and HSCs alone or in co-culture. Then, 100 μg from protein lysate was separated into a 10% SDS-PAGE gel. Proteins were transferred onto PVDF (Bio-Rad Laboratories, USA) membrane. Membranes were then blocked with 5% skimmed milk and 0.05% Tween 20 in PBS (TPBS). Primary mouse anti-Cx-43 (1 µg/mL, Sigma-Aldrich, USA) and rabbit anti-N-Cad (1 µg/mL, Life Technologies, Austin, TX, USA) antibodies were added for overnight at 4 °C. Anti-GAPDH antibody (1 µg/mL, Sigma-Aldrich, USA) was used as a housekeeping control. Blots were then incubated with adequate secondary IgG antibodies (anti-mouse or anti-rabbit). Bands were visualized using ChemiDoc MP Imaging System-Bio-Rad (Bio-Rad Laboratories, USA). The intensity of bands, in the linear range of intensity, was quantified using ImageJ software (version 1.54f, U.S. National Institutes of Health, Bethesda, MD, USA).

### 2.6. Immunofluorescence Microscopy

MSCs and CD34^+^ HSCs or AML cell lines were co-cultured on coverslips. Adherent CD34^+^ HSCs or AML cells were carefully fixed with 4% paraformaldehyde. Cells were then washed with PBS, permeabilized using Triton X-100 (0.5%) and further blocked with 5% normal goat serum (NGS) in PBS for 1 h in a humidified chamber. Incubation with primary antibodies Cx-43, N-Cad, and alkaline phosphatase (2 µg/mL) diluted in 1% NGS was performed overnight at 4 °C. Cells were next washed and incubated by IgG-conjugated secondary antibodies Texas Red (anti-rabbit) and Alexa 488 (anti-mouse) (1 µg/mL) for 1 h at room temperature. Cell nuclei were stained using Hoechst 33,342 (Molecular probes) for 10 min in the dark. Cells were washed with PBS, mounted on slides using a Prolong anti-fade kit, and observed under confocal microscopy (Laser scanning confocal microscope, LSM 710, Carl Zeiss, Germany). Data analyses were performed using Zeiss Zen software (version 3.3 blue edition).

### 2.7. Duo-Link In Situ Proximity Ligation Assay

MSCs grown on glass coverslips were co-cultured with or without CD34^+^ HSCs for 24 h at 37 °C. Cells were fixed with 4% paraformaldehyde in PBS at room temperature, permeabilized with 0.1% Triton X-100 for 15 min, and then blocked in 1 h blocking stock. After incubation at room temperature with primary antibodies (anti-Cx-43 and anti-N-Cad) for 1 h and secondary antibody conjugated with oligonucleotide for 2 h, assays were performed according to the manufacturer’s instructions (Olink Bioscience, Uppsala, Sweden). Slides were imaged using confocal microscopy (Laser scanning confocal microscope, LSM 710, Carl Zeiss, Oberkochen, Germany).

### 2.8. Functional Assay of Adhesion and Communication

The dye transfer assay was performed using the membrane-permeable dye Calcein AM (Molecular Probes, Eugene, OR, USA). Upon entry of the dye into the cell, intracellular esterases rapidly cleave the molecule to the fluorescent membrane-impermeable but gap junction-permeable acid form. Briefly, MSCs were labeled with 2 μM of calcein-AM in a complete medium for 1 h at 37 °C. Then, labeled cells were incubated in serum-free medium for 30 min at 37 °C to allow any non-de-esterified dye to leave the cells and were used immediately in dye transfer experiments. Next, CD34^+^ HSCs or AML cell lines were seeded on top of labeled monolayers of MSCs for 3 h at 37 °C. Finally, suspension cells were removed, and adherent cells were washed and detached by trypsinization. Both cell populations were fixed in 4% formaldehyde and analyzed by flow cytometry (BD, Franklin Lakes, NJ, USA).

### 2.9. Osteogenic Differentiation Engagement

MSCs were seeded in 6-well plates (70 × 10^3^/well) or 12-well plates (35 × 10^3^/well) until reaching 80% confluency. Cells were either co-cultured with CD34^+^ HSCs or AML cell lines for 24 h, 21 days, or were left in a medium throughout the experiment. Osteogenic engagement medium was added to the above conditions consisting of IMDM supplemented with 250 nM dexamethasone (Sigma-Aldrich, Burlington, MA, USA). MSCs were treated for 21 days, and the medium was changed every 3 days. After 21 days, cells were harvested for the assessment of alkaline phosphatase levels by immunofluorescence and alizarin red staining.

### 2.10. Alizarin Red Staining

MSCs were seeded into 12-well plates and cultured in the presence/absence of CD34^+^ HSCs/AML cell lines (THP-1 and MOLM-13) or dexamethasone for 24 h or 21 days. Cells were washed with PBS and fixed with 4% paraformaldehyde for 20 min. Cells were stained using the Alizarin Red S reagent (40 mM) for 40 min and then washed with H_2_O. Images were taken using light microscopy.

## 3. Results

### 3.1. Reciprocal Direct Interaction Between MSCs and CD34^+^ HSCs

Mesenchymal stem cells are precursors of the BM cellular components. Therefore, MSCs monolayer interacting with CD34^+^ HSCs provide a simple and suitable in vitro model system for the stem cell niche [14,33]. The reciprocal interaction of MSCs with CD34^+^ HSCs was assessed in a co-culture system that reflects direct cell–cell contact between the two cell types (Figure 1A). CD34^+^ HSCs were directly co-cultured with MSCs at a 1:1 ratio [14,33]. After 24 h, an (~80%) population of CD34^+^ HSCs adhered to the monolayer of MSCs (adherent fraction), while some cells (~20%) remained suspended (suspension fraction). The expression of cellular adhesion and communication markers was assessed by qPCR in CD34^+^ HSCs (adherent vs. suspension fractions) as well as in MSCs (alone vs. MSC + CD34^+^ HSCs). After 24 h of direct co-culture, the expression of stemness markers such as Oct-4 and Nanog exhibited a decrease of 28.89 ± 2.95% (*p* = 0.0006) and 27.71 ± 7.04% (*p* = 0.0171), respectively, in MSCs, suggesting that the sub-population of CD34^+^ HSCs affect their self-renewal properties and may induce MSCs differentiation. Furthermore, the expression of adhesion markers, specifically N-Cad, harbors 17.21 ± 2.39% (*p* = 0.0020) of a significant increase in MSCs (Figure 1B). The up-regulation in the expression of Cx-43 (66.25 ± 13.70, *p* < 0.0001) and N-Cad (173 ± 36.52, *p* < 0.0001) was further confirmed at the protein level (Figure 1C). It is noteworthy that these two proteins were among the most expressed proteins in MSCs (Appendix A).

The communication and adhesion markers Cx-43 and N-Cad were 88.64 ± 45.39%, *p* = 0.1226 and 88.46 ± 30.52%, *p* = 0.0442 up-regulated in the adherent fraction of HSCs, respectively (Figure 1D). Interestingly, the mRNA expression of CXCR4 implicated in homing to the bone marrow and in HSC quiescence was 95.5 ± 18.7% (*p* = 0.0070) up-regulated in HSC adherent fraction as well as the angiogenic gene VEGF (44.43 ± 9.21%, *p* = 0.0085 of increase). CFU assay was performed to test the ability of CD34^+^ HSCs to undergo terminal differentiation in cytokine-supplemented methylcellulose medium. The number of colonies formed by the adherent fraction of CD34^+^ HSCs was increased when compared to CD34^+^ HSCs cultured alone (Figure 1E). Interestingly, the percentage of CD45^+^ cells (a differentiation marker in mature lymphohematopoietic cells) was more pronounced (~4 folds) in HSCs’ suspension fraction (66.6%) when compared to the adherent fraction (13.6%) (Figure 1F), thus suggesting that the communication between MSCs and CD34^+^ HSCs conserves the stemness of CD34^+^ HSCs.

Cx-43 and N-Cad interaction is the major player in cell–cell communication through gap junctional complex. After 24 h of direct co-culture, Cx-43 and N-Cad were up-regulated in both MSCs and CD34^+^ HSCs, suggesting that cell–cell communication might be mediated through gap junctional complexes. To investigate the interaction between Cx-43 and N-Cad, the expression and localization of these two proteins in CD34^+^ HSCs and MSCs were examined using immunofluorescence microscopy (Figure 2A). Interestingly, Cx-43 and N-Cad were detected and specifically localized at the cell membrane level, in the zone of contact between adjacent MSCs and CD34^+^ HSCs. The interaction between Cx-43 and N-Cad was further asserted using the proximity ligation assay in MSCs cultured with CD34^+^ HSCs (Figure 2B). These findings are consistent with the observed co-localization of Cx-43 and N-Cad proteins by immunofluorescence and the co-immunoprecipitation assay performed with both proteins (Appendix A).

Whether Cx-43 and N-Cad assembly form a functional gap junctional complex was then investigated using the Calcein dye transfer assay. Briefly, MSCs were labeled with Calcein-AM dye and then co-cultured for 3 h with unlabeled CD34^+^ HSCs. Dye transfer between both cell types was evaluated by flow cytometry and mean fluorescence intensity (MFI) was quantified. Interestingly, a shift in MFI was noticed only in the HSCs’ adherent fraction (Figure 2C), demonstrating that cell–cell communication between MSCs and HSCs is mediated through functional gap junctional complexes, which are formed as early as 3 h.

Based on the aforementioned results, the expression of stemness markers was decreased in MSCs while the clonogenic potential of CD34^+^ HSCs was increased, suggesting that CD34^+^ HSCs self-renewal properties were affected and that MSCs differentiation may be hindered. To investigate whether the differentiation of MSCs is affected by direct communication with CD34^+^ HSCs, an engagement into osteoblastic differentiation is carried out on MSCs having direct contact with HSCs for 24 h or 21 days using dexamethasone. Alkaline phosphatase protein (ALP), an early osteoblastic marker, was expressed only in MSCs that were in contact with CD34^+^ HSCs for a short period of time (24 h). A longer co-culture period of 21 days, resulted in the failure of MSCs to be engaged in differentiation as demonstrated by low mRNA expression of ALP and high expression of Oct-4 (stemness marker) (Figure 3A,B). These results were confirmed by Alizarin red staining (Figure 3A). MSCs were more engaged in osteoblastic differentiation, only after the removal of CD34^+^ HSCs. Hence, the sustained intercellular communication between HSCs and MSCs seems to conserve the stemness of MSCs.

### 3.2. Direct Cell–Cell Communication Is Decreased in Acute Myeloid Leukemia

We previously showed that AML primary cells and cell lines have lower expression of Cx-43 [17]. We interrogated whether the communication of MSCs is modulated under pathological conditions using the same co-culture model of MSCs in direct contact with two AML cell lines: THP-1 and Molm-13 (Figure 4A). Our results demonstrated that the mRNA expression of Cx-43 and N-Cad was significantly decreased in MSCs after 24 h of co-culture with THP-1 (58% for Cx-43, *p* = 0.0028 and 55% for N-Cad, *p* = 0.0003) and Molm-13 (61% for Cx-43, *p* = 0.0138 and 70% for N-Cad, *p* < 0.0001) (Figure 4B). These results were further confirmed at the protein levels, where the expression of Cx-43 decreased to 43.7 ± 18.55% (*p* = 0.0781) in THP-1 and to 40.59 ± 16.64% (*p* = 0.0926) in Molm-13 as well as the expression of N-Cad decreased to 64.79 ± 6.87% (*p* = 0.0007) in THP-1 and to 84 ± 3.095% (*p* = 0.0001) in Molm-13 cell lines. Furthermore, using the Calcein dye transfer assay, we demonstrated that the MFI shift was evident in the adherent fraction of THP-1 and Molm-13 after co-culture (Figure 4C). Consequently, despite a lower number of forming junctional complexes between AML cell lines and MSCs, the functionality is preserved. Finally, to assess the differentiation potential of MSCs following their direct contact with AML cell lines, we co-cultured THP-1 and Molm-13 for 24 h or 21 days, in the presence of dexamethasone, a compound known for inducing osteoblastic differentiation. Our results showed that MSCs lose their differentiation ability, only when communication with AML cell lines persists for a longer period (21 days) (Figure 4D).

To further assess the regulation of cell–cell communication in leukemia patients, we performed a co-culture of MSCs with AML primary cells isolated from 3 different patients. Consistent with the obtained results with AML cell lines, Cx-43 and N-Cad protein levels were diminished in MSCs to 35.31 ± 11.37%, *p* = 0.0360 and 37.87 ± 13.71%, *p* = 0.0507, respectively. These results suggest a reduced formation of gap junction, at least with Cx-43 as a junctional partner (Figure 4E).

Finally, to explore whether BM-MSCs create a lower gap junction assembly, we interchanged the source of MSCs, either from two healthy individuals (one from an adult and one from an infant) or from two AML patients and performed the co-culture with CD34^+^ HSC (isolated from healthy individuals) and CD34^+^ AML (isolated from AML patients). Following co-culture, the expression levels of Cx-43 and N-Cad proteins were more pronounced in healthy MSCs in comparison to AML-MSCs, regardless of whether the source of CD34^+^ was from normal or AML patients (Figure 4F and Appendix A). These results suggest that the gap junction formation is reduced, probably due to a pathological defect, mainly in MSCs and not CD34^+^ cells.

## 4. Discussion

The cellular players of the hematopoietic niche direct self-renewal and differentiation of HSCs under physiological conditions. Using an in vitro co-culture model, we showed that the direct contact between CD34^+^ HSCs and MSCs is mediated by junctional complexes, including Cx-43 and N-Cad proteins as main players. We also demonstrated that the stemness steady-state situation of MSCs is preserved through direct communication with CD34^+^ HSCs, which determines the fate of both cell types. Under pathological conditions such as AML, we validated that the gap junction formation is reduced due to a decrease in Cx-43 and N-Cad expressions. We suggested that MSCs isolated from AML patients could be the source of gap junction impairment.

The microenvironment regulates the fate of HSC through direct or indirect communication, enabling the generation of all blood cells. Few studies addressed the role of direct communication in affecting the properties of MSCs and HSCs [7,14]. The interaction between both cell types regulates the hematopoietic process on two different levels. First, the adhesion receptors and their ligands, expressed on the surface of both cell types, mediate either MSCs-HSCs or HSCs-ECM adhesion. These receptors and ligands function as signaling molecules capable of altering the proliferative behavior of HSCs [14]. Second, direct intercellular communication via gap junctions could allow the exchange of low molecular weight regulatory molecules between these cells [2,5,14,34]. We showed that under normal physiological conditions, MSCs directly communicate with CD34^+^ HSCs and thus modulate the fate of both counterparts. We also confirmed that the expression of Cx-43 and N-Cad was up-regulated following the direct interaction of MSCs and CD34^+^ HSCs. The co-localization of Cx-43 and N-Cad highly suggests that one of the modalities of cell–cell communication between MSCs and CD34^+^ HSCs is facilitated through junctional complexes. In agreement with our study, it was previously demonstrated that functional Cx-43-mediated direct gap junctional intercellular communication between BM stromal cells and between stromal and hematopoietic cells is required for hematopoiesis in vitro and in vivo [35,36,37]. In fact, Cx-43 was shown to have a protective role on HSCs during hematopoietic regeneration where Cx-43 mediated the transfer of reactive oxygen species from HSCs to the hematopoietic microenvironment, thus preventing their quiescence/senescence [38].

Although the precise cellular and molecular composition of endosteal HSC niches is not fully described, the close proximity of MSCs-HSCs forms a reticular network that decides the fate of both cell types. It has been shown that the expansion and maintenance of HSCs ex vivo requires the presence of MSCs as feeders that are capable of simultaneously preserving HSC’s stemness [39]. Moreover, in the BM niche, a cooperative regulation among cytokine signals and adhesion molecules is required for the maintenance of HSCs. Our results demonstrated that following direct contact with MSCs-CD34^+^ HSCs, not only was the clonogenic potential of HSCs increased, but the expression of stemness markers in MSCs was also decreased. This finding points to the reciprocal benefits of both cells provided by direct cell–cell contact. In the same context, we proved that the cellular junctional complex delays MSCs osteoblastic differentiation. Despite the presence of dexamethasone as an exogenous signal, MSCs failed to be engaged in osteoblast differentiation when they were directly and sustainably communicating with CD34^+^ HSCs. The functional relevance of the crosstalk between MSCs-HSCs in the differentiation process allows the exchange of signaling molecules that hinder MSCs differentiation and impede the transition of HSCs from dormant to active state. Therefore, direct cell–cell communication might constitute an additional “stem warden” that could preserve the stemness capacity of both cell types. Whether adipogenic and chondrogenic differentiations would be altered in the same direction remains to be studied. In the presence of external signals, such as ROS, CXCL12, VEGF, GM-CSF and CXCL8, the junctional complexes will be dissociated to facilitate the mobilization of HSCs into the circulation and their differentiation into multipotent progenitor cells [9]. Consequently, free MSCs will be engaged in osteogenic/adipogenic/chondrogenic differentiation cascades, mainly under the control of BMP2 and BMP6 (bone morphogenetic proteins) [40], β-catenin dependent/independent Wnt signaling, Hedgehog, NELL1 (NEL-like protein 1, protein kinase C-binding protein), and IGF signaling [41].

The genesis of a leukemic permissive niche was shown to be controlled by various alterations in cell adhesion and cytokine signaling of BM-derived stromal cells, specifically MSCs [42,43]. In this study, we showed that under pathological conditions, gap junction communication through Cx-43 and N-Cad between MSCs and AML cell lines was still persistent albeit to a lesser extent. Whether lower gap junction formation depends on a signal initiated from cancer cells toward MSCs needs to be further investigated. These results are in agreement with our previous results confirming that leukemic cells isolated from AML patients lose their Cx-43 profile in comparison to normal BM-CD34^+^ cells and overexpress numerous connexins, especially Cx-25 [17]. Consequently, Cx-25—or any other connexins—could be more actively involved in gap junctional formation than Cx-43 under leukemic conditions. Additional investigations are needed to confirm the involvement of other connexins in our co-culture model. In line with our data, carbenoxolone (CBX), a gap junction disruptor, shows antileukemic effects against AML cell lines by triggering apoptosis without affecting normal BM-CD34^+^ cells. Furthermore, in a xenogenic human AML model, CBX treatment improves mice survival by preventing leukemic infiltration to the liver and the spleen of treated animals [17].

The tumor microenvironment plays a major role in cancer progression by providing nutrients and survival signals to tumor cells [44]. This microenvironment protects cancer cells from normal immune responses and therapeutic treatment agents [45]. We demonstrated that a direct co-culture between normal MSCs and AML cell lines, or AML primary cells isolated from patients, induces a down-regulation of the expression of Cx-43 and N-Cad, lowering their capacity to form junctional complexes. Since our data suggest that AML-MSCs have low Cx-43 levels, the direct communication of MSCs with other cell types in the BM niche could be compromised as well. Moreover, the differentiation ability of MSCs was abolished after a long nesting time with AML cell lines, further pointing to the role of communication in the regulation of leukemic cell proliferation and chemoresistance [46]. In the same context, previous studies showed that Cx-43-junctional activity increased multiple myeloma cell survival and chemoresistance when in contact with stromal cells [47].

## 5. Conclusions

In conclusion, we propose that under physiological conditions, MSCs-CD34^+^ HSCs direct communication preserves the stemness state of both cells via gap junctions, a mechanism that is possibly altered in a leukemic microenvironment. The switch to using other connexins in cell–cell communication might be another pathway undertaken for leukemogenesis.

## Figures and Tables

**Figure 1 cancers-16-03972-f001:**
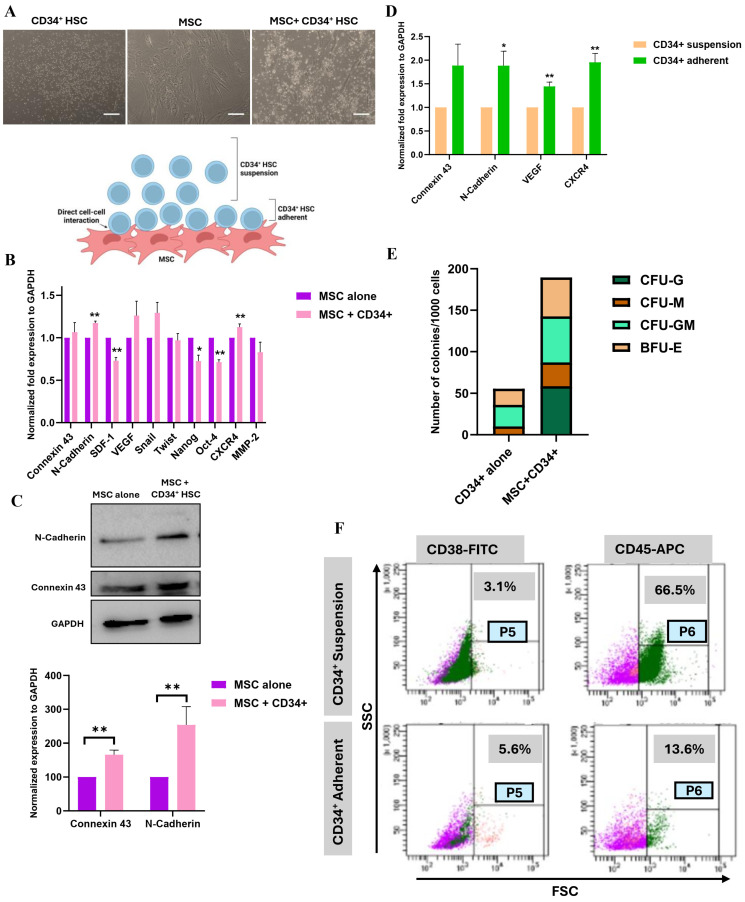
Direct co-culture of MSCs and CD34^+^ HSCs: (**A**) Schematic representation of the direct co-culture system between MSCs and CD34^+^ HSCs. Representative images of HSCs alone, MSCs alone, and MSCs + CD34^+^ HSCs after direct co-culture. Scale bar, 100 µm. (**B**) Histogram representing mRNA expression levels of adhesion and communication markers in MSCs following direct co-culture, assessed by qPCR. * *p* ≤ 0.05 and ** *p* ≤ 0.01 (*t*-test). (**C**) Western blot of Cx-43 and N-Cad expression in MSCs after 24 h of direct co-culture with CD34^+^ HSCs. Results are represented as normalized expression to GAPDH in three independent experiments ± SEM. ** *p* ≤ 0.01 (*t*-test). (**D**) Expression of adhesion/communication markers and VEGF, CXCR4 in CD34^+^ HSCs following direct co-culture, assessed by qPCR. Results are represented as normalized expression of GAPDH in three independent experiments. * *p* ≤ 0.05 and ** *p* ≤ 0.01 (*t*-test). (**E**) Clonogenic potential of CD34^+^ HSCs, expressed as the number of colonies after direct co-culture with MSCs. (**F**) Cell profiling of CD34^+^ HSCs after co-culture, showing the percentage of CD38^+^ and CD45^+^ cells in both suspension and adherent fractions. Green population refers to CD45^+^ cells, red population refers to CD38^+^ cells and purple population refers to CD45^−^CD38^−^ cells.

**Figure 2 cancers-16-03972-f002:**
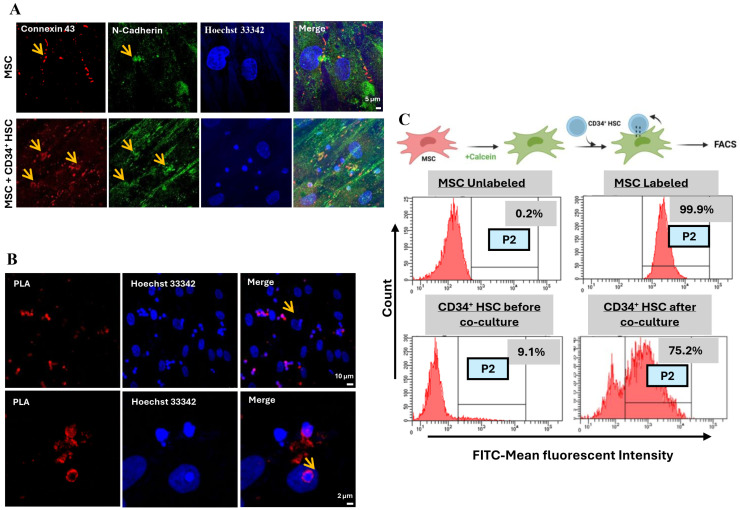
(**A**) Double immunostaining of Cx-43 (red), and N-Cad (green) in MSC alone and MSC + CD34^+^ HSC following direct co-culture. Arrows point to the co-localization of the two proteins. Scale bar, 5 µm. (**B**) Cx-43 and N-Cad interaction in MSCs and HSCs after direct co-culture as detected by Duo-Link assay. The dots (red) are representatives of the close proximity of the two proteins of interest. Nuclei were stained with Hoechst 33,342 dye (blue). Scale bar, 10 µm (upper panel) and 2 µm (lower panel). (**C**) Representative flow cytometry graph showing the shift in MFI following co-culture of unlabeled HSCs with Calcein-labeled MSCs.

**Figure 3 cancers-16-03972-f003:**
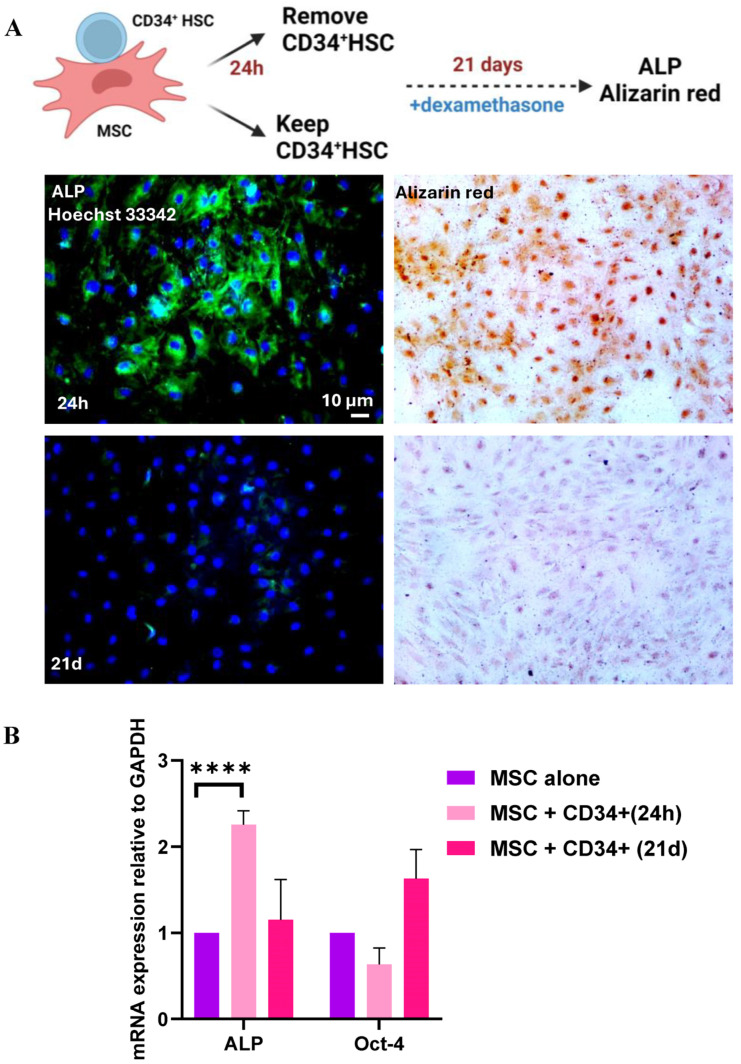
Engagement in osteoblastic differentiation of MSCs after co-culture with HSCs for 24 h or 21 days: (**A**) Left panel: Expression of ALP protein in MSCs detected by immunofluorescence. Right panel: Alizarin red staining of MSCs visualized by light microscopy. Scale bar, 20 µm. (**B**) mRNA expression of ALP and Oct-4 in MSC following direct interaction with CD34^+^ HSC for 24 h and 21 d. **** *p* ≤ 0.0001 (*t*-test).

**Figure 4 cancers-16-03972-f004:**
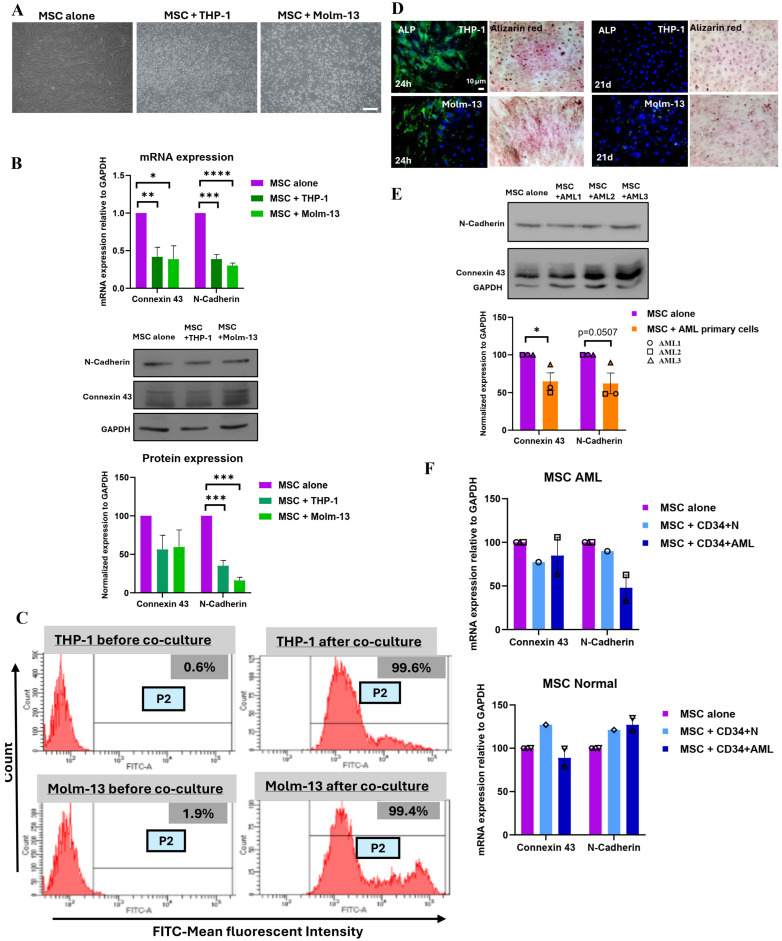
Direct co-culture of MSCs and AML cells: (**A**) Schematic representation of the direct co-culture system between MSCs and THP-1 and Molm-13. Scale bar, 100 µm. (**B**) mRNA and protein expression levels of Cx-43 and N-Cad in MSCs following direct co-culture with AML cell lines, assessed by qPCR and Western blot, respectively. * *p* ≤ 0.05, ** *p* ≤ 0.01, *** *p* ≤ 0.001, **** *p* ≤ 0.0001 (*t*-test). (**C**) Representative flow cytometry graphs showing the shift in MFI following co-culture of unlabeled AML cell lines (THP-1 and Molm-13) with Calcein-labeled MSCs. (**D**) Fluorescent and light microscopy images of MSCs co-cultured with AML cell lines for ALP expression and Alizarin Red staining, respectively. Scale bar, 20 µm. (**E**) Expression of Cx-43 and N-Cad in MSCs following direct co-culture with AML primary cells (3 patients), assessed by Western blot analysis. * *p* ≤ 0.05 (*t*-test). (**F**) mRNA expression levels of Cx-43 and N-Cad in MSCs of healthy and AML patients following direct co-culture with AML cell lines, CD34^+^ N from healthy individuals (*n* = 1) and CD34^+^ AML from patients (*n* = 2).

**Table 1 cancers-16-03972-t001:** List of human primers used for qPCR.

**Genes**	**Sequence**
Connexin 43	F: GGA AGA TGG GCT CAT GAA AA R: GCA AAG GCC TGT AAC ACC AT
N-Cadherin	F: GGT GGA GGA GAA GAA GAC CAG R: GGC ATC AGG CTC CAC AGT
MMP2	F: TTG ACG GTA AGG ACG GAC TC R: ACT TGC AGT ACT CCC CAT CG
SDF-1	F: GCC CGT CAG CCT GAG CTA CA R: TTC TTC AGC CGG CGT ACA ATC T
VEGF	F: AGG CCC ACA GGG ATT TTC TT R: ATC AAA CCT CAC CAA GGC CA
Snail	F: CTT CCA GCA GCC CTA CGA C R: CGG TGG GGT TGA GGA TCT
Twist	F: AGC TAC GCC TTC TCG GTC T R: CCT TCT CTG GAA ACA ATG ACA TC
Nanog	F: CAG AAG GCC TCA GCA CCT AC R: ATT GTT CCA GGT CTG GTT GC
Oct-4	F: CAG TGC CCG AAA CCC ACA C R: GGA GGAC CCA GCA GCC TCA AA
CXCR4	F: CCT CCT GCT GAC TAT TCC CGA R: GGA ACA CAA CCA CCC ACA AGT
ALP	F: GCTGTAAGGACATCGCCTACCA R: CCTGGCTTTCTCGTCACTCTCA
GAPDH	F: TGG TGC TCA GTG TAG CCC AG R: GGA CCT TGA CCT GCC GTC TAG

## Data Availability

No new data were created or analyzed in this study. Data sharing is not applicable to this article.

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
