# Peer review of "Direct Interaction Between CD34+ Hematopoietic Stem Cells and Mesenchymal Stem Cells Reciprocally Preserves Stemness"

_cancers, 2024, doi:10.3390/cancers16233972_

Round 1

Reviewer 1 Report

Comments and Suggestions for Authors

The authors Safi et al have studied cell cell communication between HSCs and MSCs using a co culture model system. The authors suggests that connexin-43 and N-cadherin interaction preserves stemness of both CD34+ HSCs and MSCs.Additionally they observed the healthy MSCs have an higher expression of N- cadherin and Connexin -43 compared to AML MSC irrespective of the source of the CD34+ cells. While the authors have tapped into an interesting aspect of AML biology that could drive resistance, I think that addressing the following questions will greatly improve the study.

Minor comments:

1.    Page 2, line 45: Grammatical errors

2.    Page 2, line 68: Grammatical error. Should be membrane proteins that mediate exchange of small molecules.

3.    Page 2, line 92-93: The chemoresistance is induced by BM derived MSCs in the AML cells. This sentence needs to be rephrased as the message is not clear in the current format.

4.    Page 3, Line 138: Details regarding AML growth media needs to be included.

5.    Page 4: For the CFU assay the authors have used IMDM media for CFU assay? Can the authors cross check this? CFUs are typically done in methylcellulose media.

6.    Page 4: Was EPO added to the media to obtain erythroid colonies? If cytokines were supplemented in the media the concentration for each cytokine need.

7.    Maintain consistency with the colors for each set. MSC and MSC alone colors should be consistent across figures.

8.    Include the sex and other necessary details of the CD34+ HSCs used in the assays.

9.    Fig 4A MSC alone has not SD or SEM error bars.

10. Figure 4C label the MFI for the dye that is being measured.

11. Fig 4F label the axes for the gene whose expression is being checked.

12. Include the FAB subtyme of the primary AML patients.

13. Are the AML patients similar in their FAB subtype as that of THP1 and MOLM 13?

Major comments:

1.    The authors claim that N-cadherin and Cx-43 are essential for cell cell communication. They have shown that there is upregulation of these proteins in both the MSCs and HSCs. However, it is not clear from the experiments to understand the contribution of the cell type that maybe essential for the colony forming potential of the HSCs. In this context the authors need to KD both N-cadherin and Cx-43 in the MSCs and HSCs to determine if the CFU potential of the HSCs are affected in absence of expression of N-cadherin or Cx-43 in the MSCs. Alternatively, pre-treatment of the MSCs with blockers for N-cadherin and Cx-43 can be used before co culturing with CD34+ HSCs to reach a reasonable conclusion.

2.    Figure 2A it is not clear which cell is expressing N-cadherin and which is expressing Cx-43. The authors need to include cell surface markers for cell type to differentiate between the MSC and HSCs in their IF assays.

3.    For figure 3 The authors claimed that co culture with CD34+ HSC affect the stemness of the MSCs and induce MSC differentiation. Can the authors check if expression of osteoblastic transcription factors like Osx, and Ocn are altered during short term and long term co culture assay?

4.    MSCs are capable of trilineage differentiation. Is MSC differentiation induced only along the osteoblastic direction. It might be worthwhile to check for expression of transcription factors and marker like PPARg and Sox4 that are expressed during adipogeneic and chondrogenic differentiation.

5.    The change in protein levels of Cx-43 and N-cadherin after co culture is not evident in the WB Figure 4B. The authors can provide higher or lower exposure blots to confirm their claims.

6.    Figure 4F are the results statistically significant?

7.    Figure 4F was the co culture a short term co culture 24 hrs or a long term co culture for 21 days followed by differentiation?

Comments on the Quality of English Language

Grammatical errors pointed out in minor comments.

Reviewer 2 Report

Comments and Suggestions for Authors

The interaction between CD34+HSCs and MSCs is known to preserve stemness, and its alteration in pathologic conditions contributes to leukemia dissemination.

In this paper the authors performed a series of elegant in vitro experiments to clarify the relationship between MSCs and normal CD34+HSC, AML cell-lines or primary AML cells. 

They  demonstrated that in AML direct cell-cell communication was decreased as well the ability to form gap-junctions. Moreover they observed that the impaired gap-junction assembly was ascribable to MSCs, irrespective to the source of co-cultured CD34+ cells.

The paper is interesting and adds some new information of the complex organization of BM microenvironment. Experiments and figure are well explained, references updated.

I have just few questions:

·      Could the reduction of cell-cell interaction in AML (cell-lines and primary) depend to the lower stemness of AML bulky population compared to normal CD34+HSCs?

·      How did they explain the high capacity of AML to survive in the BM microenvironment?

·      Could the MSCs altered capacity to form gap-junctions affect also MSCs communication with BM lymphocytes?  

Round 2

Reviewer 1 Report

Comments and Suggestions for Authors

All the concerns have been satisfactorily addressed by the authors in the revision.